# The Association between Childhood Immunization and Gender Inequality: A Multi-Country Ecological Analysis of Zero-Dose DTP Prevalence and DTP3 Immunization Coverage

**DOI:** 10.3390/vaccines10071032

**Published:** 2022-06-27

**Authors:** Cecilia Vidal Fuertes, Nicole E. Johns, Tracey S. Goodman, Shirin Heidari, Jean Munro, Ahmad Reza Hosseinpoor

**Affiliations:** 1Department of Data and Analytics, World Health Organization, 20 Avenue Appia, 1211 Geneva, Switzerland; vidalc@who.int (C.V.F.); johnsn@who.int (N.E.J.); 2Department of Immunization, Vaccines, and Biologicals, World Health Organization, 20 Avenue Appia, 1211 Geneva, Switzerland; goodmant@who.int (T.S.G.); heidaris@who.int (S.H.); 3Gavi, The Vaccine Alliance, Chemin du Pommier 40, Le Grand-Saconnex, 1218 Geneva, Switzerland; jmunro@gavi.org

**Keywords:** immunization, vaccination, zero-dose children, diphtheria-tetanus-pertussis vaccine, determinants of immunization, health status disparities, gender equity

## Abstract

This study explores the association between childhood immunization and gender inequality at the national level. Data for the study include annual country-level estimates of immunization among children aged 12–23 months, indicators of gender inequality, and associated factors for up to 165 countries from 2010–2019. The study examined the association between gender inequality, as measured by the gender development index and the gender inequality index, and two key outcomes: prevalence of children who received no doses of the DTP vaccine (zero-dose children) and children who received the third dose of the DTP vaccine (DTP3 coverage). Unadjusted and adjusted fractional logit regression models were used to identify the association between immunization and gender inequality. Gender inequality, as measured by the Gender Development Index, was positively and significantly associated with the proportion of zero-dose children (high inequality AOR = 1.61, 95% CI: 1.13–2.30). Consistently, full DTP3 immunization was negatively and significantly associated with gender inequality (high inequality AOR = 0.63, 95% CI: 0.46–0.86). These associations were robust to the use of an alternative gender inequality measure (the Gender Inequality Index) and were consistent across a range of model specifications controlling for demographic, economic, education, and health-related factors. Gender inequality at the national level is predictive of childhood immunization coverage, highlighting that addressing gender barriers is imperative to achieve universal coverage in immunization and to ensure that no child is left behind in routine vaccination.

## 1. Introduction

Gender equality is not only an important standalone goal, but also a key contributor to and indicator of health of populations more broadly [1]. Gender inequality has been linked to greater mortality [2,3,4] and morbidity across a number of health outcomes, such as heart disease [5], HIV [6], TB [7], and others. In the field of childhood immunization, gender-related barriers are increasingly recognized as meaningful drivers of persistent low immunization rates and inequalities in immunization coverage [8,9,10]. These gender-related barriers include a range of factors, such as lack of access to education, limited household healthcare decision-making, gender-based violence, and restricted mobility [11,12]. Gender inequality contributes to each of these barriers, and can be considered a barrier to care itself [10]. Though work has examined differences in immunization coverage rates by sex [13,14], the relationship between childhood immunization coverage and gender inequality in a population more broadly has received less attention.

In addition to gender inequality in specific health outcomes or determinants, aggregate measures of gender inequality more broadly can be informative indicators that influence population health. Two widely-used measures of gender inequality at the population level are the Gender Development Index (GDI) [15] and the Gender Inequality Index (GII) [16]. The GDI measures gender inequalities in achievement in three basic dimensions of human development (health, education, and control over economic resources), while the GII measures gender-based advantage or disadvantage in reproductive health, empowerment, and the labor market. These measures may be correlated with health outcomes directly through their component items (such as female education), or more proximally through the associated gender norms, policies, and institutions which those aggregate measures of gender inequality reflect. Gender inequality as measured by these indices has been shown to be associated with child health outcomes including mortality rates [2] and immunization coverage [17,18]. Only one study to date has examined gender inequality and childhood immunization coverage at the national level across 45 countries using data from 2005–2014, and found that greater gender inequality was associated with lower and less equitable immunization coverage [17].

To examine the relationship between gender inequality indices and childhood immunization across countries, we utilize two outcomes related to the combined diphtheria, tetanus and pertussis vaccine (DTP): zero-dose children, or zero-dose DTP, a proxy for children who have missed immunization services entirely, and DTP3 immunization coverage (DTP3), a proxy for children who have accessed the full series of basic immunizations. These outcomes together represent both extremes of the immunization cascade [19]. The DTP vaccine was first developed in 1948, and was among the initial vaccines included in the World Health Organization’s (WHO) Expanded Programme on Immunization upon its founding in 1974 [20,21]. Today, DTP vaccines exist in various forms, including DTaP/Tdap and pentavalent vaccine, which also includes protection against Hepatitis B and Hib. DTP vaccines are generally given as a three-dose series four weeks apart at 6 weeks, 10 weeks, and 14 weeks of age [22]. DTP3 coverage is frequently used as an indicator of child health and of health system function and performance broadly, as it requires regular and timely interaction with routine health systems [23,24]. While DTP3 coverage remains a standard indicator globally, the prevalence of zero-dose children is an equally important indicator of immunization service equity [25]. Major global immunization initiatives including the Immunization Agenda 2030 and the Gavi Phase 5 strategy feature the theme of leaving no one behind, highlighting a need to identify children who do not receive immunizations and understand factors associated with immunization non-receipt [26,27]. A wide range of factors are known to be associated with non- or under-vaccination, including poverty [25,28], remote rural residence [29], conflict [30], migration [31], homelessness, cultural marginalization, and, importantly, gender-related barriers [32]. Centering gender equity and considering gender-related factors in childhood immunization activities is, thus, crucial to ensure that no children are left behind.

In this study, we assess whether gender inequality is related to immunization at the national level. We expand upon previous work in this area [17] using recent data, multiple measures of gender inequality, and a more globally representative sample of countries (e.g., including low, middle, and high-income countries). Our objective is to evaluate whether there is a significant relationship between national gender inequality and immunization coverage, using an ecological analysis approach. We hypothesize that gender inequality in a country, as measured by national-level GDI and GII, reflects persistent gender-related barriers faced by caregivers and guardians in accessing healthcare for their children, and therefore will be correlated with lower immunization coverage at the national level.

## 2. Materials and Methods

### 2.1. Indicators and Data Sources

The data used in this study include annual national estimates of childhood immunization, indicators of gender inequality and other demographic, economic and social characteristics. Data were available for up to 165 countries per indicator per year, spanning from 2010 through 2019.

#### 2.1.1. Immunization Outcomes

The associations between childhood immunization and gender inequality were assessed for two primary outcomes based on national coverage of the DTP vaccine among children under one. The first outcome was the prevalence of zero-dose children, or zero-dose DTP, defined as the percentage of surviving one-year old children who have not received the first dose of the DTP vaccine series. The second outcome was the prevalence of DTP3 immunization (DTP3), that is, the percentage of one-year old children who have received three doses of the DTP vaccine.

#### 2.1.2. Factors Associated with Immunization Coverage

We examined factors selected a priori to account for demographic, geographic, and other human development characteristics that have been shown in the literature to be associated with national childhood immunization levels [33]. These included average annual rate of population change, percent of population under 15 years of age, percent of population living in urban areas, and a number of human development indicators. Variables were selected to account for demand and supply side factors that influence vaccination and might confound the association between immunization and gender inequality (summary statistics can be found in Appendix A).

To capture human development, we first utilized the human development index (HDI), which is a summary measure of achievements in three key dimensions of human development: a long and healthy life, access to knowledge, and a decent standard of living. HDI is computed as the geometric mean of normalized indices for each of the three dimensions [34]. We analyzed HDI both as a single index and as the three dimension-specific indices, all normalized between 0 and 1. The health index is based on life expectancy at birth, the education index based on mean expected years of schooling for children and mean years of schooling for adults ages 25 years and older, and the income index is based on gross national income per capita (2017 purchasing power parities (PPP) in USD). We separately examined three individual non-standardized indicators which reflect the same three dimensions of human development: health expenditure per capita PPP, mean years of schooling for the population 25 or older, and GDP per capita PPP.

#### 2.1.3. Gender Inequality

Gender inequality was measured using two metrics: the gender development index (GDI) and the gender inequality index (GII). Gender inequality is a complex construct to capture quantitatively; we chose, therefore, to examine two measures to ensure robustness of findings.

GDI measures gender inequalities in achievement in the three basic dimensions of human development captured by the HDI: health, education, and control over economic resources. To calculate the GDI, the HDI is calculated separately for men and women in a country, and the GDI is the ratio of HDI value among women to HDI value among men. Additional detail on GDI construction has been published elsewhere [35].
(1)$$\mathrm{GDI}=\frac{{\mathrm{HDI}}_{w}}{{\mathrm{HDI}}_{m}}$$

GDI values below 1 indicate higher human development among men than women, a value equal to 1 indicates equality, and values above 1 indicate higher development among women than men. We created a binary analysis variable for GDI based on quintiles of its sample distribution (cutoff values for the 20th, 40th, 60th, and 80th percentiles of GDI were 0.877, 0.942, 0.971, and 0.990, respectively), dichotomized to high gender inequality favoring men (Q1) vs. medium/low/negligible gender inequality (Q2–Q5). We analyzed GDI as this binary measure in regression analyses.

GII measures gender-based disadvantage in three dimensions: reproductive health (measured by maternal mortality and adolescent birth rate), empowerment (measured by share of seats in parliament or equivalent political office and population with at least secondary education), and the labor market (measured by labor force participation). It shows the loss in potential human development due to inequality between women’s and men’s achievements in these dimensions. It is scaled from 0 to 1, with 0 reflecting a situation where women and men fare equally, and 1 reflecting a situation where one gender fares as poorly as possible in all measured dimensions [35]. We analyzed the GII as a continuous measure in regression analyses.

GDI and GII were selected as they are two widely-used gender inequality indicators which are publicly available at the national level across countries and across years.

Table 1 presents a summary of indicators and data sources [36,37,38,39,40].

### 2.2. Analyses

We first produced basic descriptive statistics and cross tabulations of immunization outcomes and gender inequality indices, as well as unadjusted outcome distributions by levels of gender inequality, for the most recent year of data available (2019). We then conducted a series of regression analyses to examine the association between childhood immunization and the level of gender inequality using a pooled 10-year dataset. All country-years for which data were available were included in analyses. As the outcomes are transformed to proportions with values between 0 and 1, all models were estimated using fractional logit models (as linear models do not ensure that the expected value is between 0 and 1) [41,42].

The outcome indicator, Y (zero-dose DTP or DTP3 coverage), was estimated as a function of an indicator of gender inequality and other covariates. The estimated fractional logit model has the form:(2)$${E(Y}_{\mathrm{it}}{|X}_{\mathrm{it}})=G\left(\theta_{t}+{\mathrm{GI}}_{\mathrm{it}}+X_{\mathrm{it}}\beta \right)$$
where i indexes country and t indexes year. GI is the measure of gender inequality, either the GDI or the GII in binary or continuous form, respectively. X is a vector that includes controls for population growth and age structure; share of urban population; and specific indicators of economic, education and health development. In addition, θt are year fixed effects (year dummies) which capture average changes in the immunization outcome over time and control for factors changing each year that are common to all countries for a given year. G(*) is the logistic cumulative density function.

Models that used GDI were estimated with a binary variable equal to 1 if countries were in the high gender inequality category (Q1), and 0 if countries were in any of the four medium/low/negligible inequality categories (Q2 to Q5). Models that used GII were estimated including the index as a continuous variable measured between 0 and 1. Summary immunization coverage levels are presented by binary (highest inequality quintile vs. not) levels of both indicators.

For each immunization outcome and measure of gender inequality, four models were estimated:Model (1) estimates the unadjusted association between the outcome and gender inequality (GDI or GII), without controlling for any other factors;Model (2) includes controls for annual population growth and age structure (measured as the percentage of the population under 15 years of age), percentage of urban population, and the three individual dimensional indices of the HDI (health index, education index, and GNI index);Model (3) includes the same controls as Model (2), but the level of human development is measured through the overall HDI, instead of the three dimension-specific indices;Model (4) includes the same controls as Model (2), but the level of human development is measured through three specific economic, education, and health indicators, namely: natural log of GDP per capita PPP, mean years of schooling for the population 25 or older, and the log of current per capita health expenditure PPP.

All four models also accounted for non-parametric time trends via year fixed effects.

All model results are presented in tables; for simplicity, we present in text and figures only results from the model with the largest likelihood and best fit, Model 2.

All regressions were estimated with standard errors clustered at the country level. Statistical significance was set at *p* < 0.05 for all comparisons including adjusted odds ratios (AORs); 95% confidence intervals (CIs) are reported throughout. All analyses were conducted using STATA 16.1 [43].

## 3. Results

### 3.1. Descriptive Analyses

Data were available for at least one year in the range 2010 to 2019 for 165 countries for GDI and for 162 countries for GII [36]. In the pooled 10-year sample, where each observation is one country-year, the mean value of GDI was 0.934, ranging from 0.482 in Yemen 2018 to 1.042 in Latvia 2010. For the GDI, a mean value below 1 indicates that, overall, human development is lower among women than men. The mean value of GII in the pooled sample was 0.364, ranging from 0.025 in Switzerland in 2019 to 0.819 in Yemen in 2015; for the GII, a mean value of 0 represents total gender equality and a value of 1 represents total inequality. Distributions of GDI and GII in 2019 can be found in Appendix A.

In 2019, the most recent year of available data, higher gender inequality was associated with higher prevalence of zero-dose DTP and lower DTP3 immunization coverage in unadjusted cross tabulations (Table 2). Countries with high gender inequality (favoring men), as measured by the GDI, had, on average, 7.5 percentage points more zero-dose prevalence (10.5% vs. 3%), and 11.5 percentage points lower DTP3 immunization coverage (82.5% vs. 94%) than countries with lower inequality. Similarly, countries with high gender inequality, as measured by the GII, had higher zero-dose prevalence (10% vs. 3%) and lower DTP3 immunization coverage (81% vs. 94%) than countries with lower inequality.

### 3.2. Regression Analyses

The results of the fractional logit regressions are presented by the measure of gender inequality used. The model with the largest likelihood and best fit (Model 2) was the preferred model to describe main results in text and figures. Full regression output for all models can be found in Appendix A.

#### 3.2.1. Gender Development Index

The first set of models are estimated using the GDI indicator, dichotomized as high gender inequality (favoring men) or not.

GDI was significantly associated with both zero-dose prevalence and DTP3 coverage in regression analyses (Table 3, Appendix A). In countries with high gender inequality, the odds of zero-dose prevalence were 1.6 times higher (AOR = 1.61, 95% CI: 1.13–2.30) compared to countries with lower inequality. Consistently, the odds of DTP3 coverage were 37% lower (AOR = 0.63, 95% CI: 0.46–0.86) in countries with high gender inequality relative to countries with lower inequality. Results are consistent and statistically significant across alternative models that control for demographic, geographic and human development characteristics.

Estimated coefficients can also be interpreted as average partial effects; that is, the percentage point change in the outcome variable (zero-dose DTP or DTP3 coverage) for a change in the category of gender inequality (See Figure 1 and Appendix A). Countries with high inequality favoring men have an expected increase of 3.1 percentage points in the prevalence of zero-dose DTP relative to countries with lower inequality, increasing from 5.8% (95% CI 4.7–6.8%) for countries with lower inequality to 8.8% (95% CI 6.6–11.1%) for countries with high inequality favoring men. A country with high gender inequality is expected to have a 4.6 percentage point lower prevalence of DTP3 immunization coverage than a country with lower inequality, dropping from 90.4% (95% CI 89.0–91.8%) for countries with lower inequality to 85.8% (95% CI 82.8–88.8%) for countries with high inequality favoring men.

#### 3.2.2. Gender Inequality Index

The association between gender inequality and childhood immunization was further examined using the GII as an alternative measure of gender inequality. The GII was analyzed as a continuous variable ranging from 0 to 1, with 0 representing total gender equality and 1 representing total gender inequality favoring men. We replicated the same four models as used to analyze GDI.

Estimated marginal effects of GII on immunization outcomes, as reported in Table 4, indicate statistically significant associations between the GII and the proportion of zero-dose children and DTP3 coverage in a country (also see Appendix A). This association remains statistically significant after accounting for demographic, geographic, and human development indicators. On average, an increase of 1 percentage point in GII is associated with a 0.17 percentage point increase (95% CI 0.06–0.28) in the percentage of zero-dose children. Conversely, gender inequality as measured by the GII has a negative and statistically significant association with the proportion of children vaccinated with three doses of the DTP vaccine. A one percentage point increase in GII is associated with a 0.25 percentage point decrease (95% CI −0.40–−0.10) in coverage of DTP3 immunization.

Figure 2 plots the average expected proportion of unvaccinated children (Panel a) and DTP3 coverage (Panel b) for fixed values of the GII across the range of observed GII values. Overall, results are consistent with those found using the GDI. For higher values of the GII, the proportion of zero-dose children increases and coverage of DTP3 decreases. Changes are larger at higher levels of inequality. These results are consistent across all model specifications.

## 4. Discussion

Results from these analyses indicate that the level of gender inequality in a country was significantly associated with national childhood immunization rates during the time period 2010–2019, with greater gender equality associated with improved immunization coverage. These findings were consistent across both examined outcomes of childhood immunization, reflecting the two extremes of the DTP immunization cascade (zero-dose DTP and DTP3), and across both examined measures of gender inequality, GDI and GII. Greater gender equality was associated with markedly better immunization coverage in unadjusted bivariate comparisons in 2019, whereby coverage of DTP3 was more than 10% higher and zero-dose prevalence was 7% lower in countries with lower gender inequality compared to countries with high gender inequality. These findings were consistent in direction and significance of effects in multi-year, multivariate regression analyses. Though precise model estimates differed, these findings were not sensitive to the choice of additional demographic and human development indicators included in regression models. In total, these findings suggest that gender inequality is a meaningful and statistically significant predictor of childhood immunization at the national level, even when accounting for other known correlates of immunization coverage including demographic, geographic, and human development indicators capturing wealth, education, and health, as well as year fixed effects which capture non-parametric time trends in both immunization and gender development.

These findings confirm calls for the reduction of gender barriers to improve immunization access, and add to the evidence that gender equality is tantamount to ensure universal coverage and equity in childhood immunization [8,10,11]. Results from this study align with a previous national-level analysis [17], despite a larger and more global sample (165 rather than 45 countries), more recent data (2010–2019 rather than 2005–2014), multiple years of data for countries (rather than a single most recent survey), and an alternate analysis approach (fractional logistic regression rather than a meta-regression approach). Robustness of the findings, despite these differences, strengthens the conclusions presented here. Results from this study also align with previous analyses examining individual-level measures of gender equity in a multi-national sample, which found that greater maternal empowerment (including gender equality in education, age at marriage, and decision-making) was associated with lower zero-dose DTP likelihood for her children [32], as well as a systematic review which found positive associations between women’s agency (including gender equality in mobility and decision-making) and DTP3 coverage [12]. Findings also align with previous qualitative work in this area; a multi-country meta-ethnographic systematic review identified gender inequality (including inequalities in education, income, autonomous decision-making, and lower social status for women generally) as a key barrier to immunization coverage [44]. Significant associations between measures of gender inequality and immunization outcomes support consideration of gender equality as a key determinant of immunization coverage, beyond consideration of differences in outcomes by child sex; while there are few differences in childhood immunization rates by sex in most examined countries [45], countries with greater gender inequality had significantly lower immunization coverage. This is an important consideration for immunization equity work and for public health more broadly.

Findings from this study suggest that policies and interventions which directly address gender inequality are needed to ensure high and equitable immunization coverage. Gender responsive measures in policy and practice can address some of the immediate challenges faced by caregivers in accessing immunization services for their children. Practical examples of such measures include establishing the location and timing of vaccine clinics based on mothers’ work schedules and time availability, holding vaccination clinics in locations that are easily accessible for mothers, or having female vaccinators in communities where it is not socially acceptable for men and women to interact. These measures can improve coverage services for a short period; however, approaches that have sustained impact will be those that address deeper inequalities, including those factors captured by the GDI and GII such as education and control of resources. Maternal education, and, by proxy, gender inequality in education, has in particular been extensively studied and shown to be a determinant of child immunization [46,47,48,49,50]. Gender transformative approaches, therefore, must focus on shifting power imbalances and addressing social norms, beliefs, and attitudes which create and sustain discriminatory policies and practices across sectors. Gender-transformative approaches can include supporting interventions that distribute household responsibilities among both parents, supporting fathers to be engaged in child care and seeking health services for children, ensuring equitable payments among health professionals, ensuring equitable access to and utilization of educational opportunities, ensuring equal representation of women and men in decision making positions, and prevention of sexual assault and harassment in the health sector [10,51]. These approaches are described in detail in the report *Why Gender Matters: Immunization Agenda 2030* [10]. Often times, these approaches must be cross-sectoral as they reach beyond the health sector. While childhood immunization, generally, is highly cost-effective [52], multi-sectoral approaches will likely not be the most cost-effective interventions short term [53]. However, systemic approaches such as these have greater potential sustained and wide-ranging impact, and key stakeholders in immunization will need to consider novel approaches such as these to vaccinate the hardest-to-reach populations in the effort to leave no one behind [26,27].

Findings from this study should be considered in light of several limitations. First, these analyses involved a cross-sectional ecological analysis, and therefore cannot demonstrate causality. However, plausible theoretical pathways between gender equality and childhood immunization outcomes, as well as prior literature with similar findings using individual-level and qualitative analyses, support the importance of the observed associations. Second, these are national-level analyses which may conceal within-country differences in association; observed associations likely underestimate the associations between gender inequality and immunization coverage among the most marginalized populations. Future work is planned to examine the relationships between measures of gender inequality and immunization coverage at the subnational level. Third, the measures of gender inequality examined here (GDI and GII) have several limitations, including capturing only some elements of the broader construct of gender inequality, and sensitivity to indicators and variable definition [54,55]. Despite these limitations, the strong and consistent association across both measures provide evidence to the association between childhood immunization and gender inequality. Future work should examine subpopulations and individual-level analyses, as well as pathways through which gender inequality can influence childhood immunization, to better understand the relationship between these factors and to best inform policy and practice.

This study also has several strengths. It uses 10 years of data from up to 165 countries, presenting a more complete global analysis than any previously published work. Findings were robust to the choice of gender inequality measure (GDI and GII) and to the choice of immunization indicator (DTP3 and zero-dose DTP), as well as to model specifications, strengthening the conclusion of significant and meaningful associations between gender inequality and coverage. Though the ecological nature of these analyses preclude any conclusions on causality, the alignment of findings with previous published work using individual-level [32], country-level [17], and qualitative data [44] collectively add strength to the argument in favor of consideration of gender barriers as key determinants to immunization coverage and equity. Future research should, in particular, implement and evaluate interventions which target gender inequality as a mechanism to improve immunization coverage, to better determine the feasibility, costs, and practical impact of gender-transformative interventions in this space. Additionally, research which examines the relative contributions of a range of barriers to immunization care would be informative for prioritization and resource allocation in vaccination efforts. Work examining measures of gender inequality in subnational populations would also add further support to the findings presented here.

## 5. Conclusions

Our study found a significant negative relationship between national gender inequality and immunization coverage, using recent data from 165 countries and an ecological analysis approach. The results produced in this study strengthen the evidence base emphasizing the negative impact of gender inequality and gender related barriers in childhood immunization coverage, and bolster calls for gender-transformative policies and practices to ensure universal childhood immunization coverage and equity in immunization services.

## Figures and Tables

**Figure 1 vaccines-10-01032-f001:**
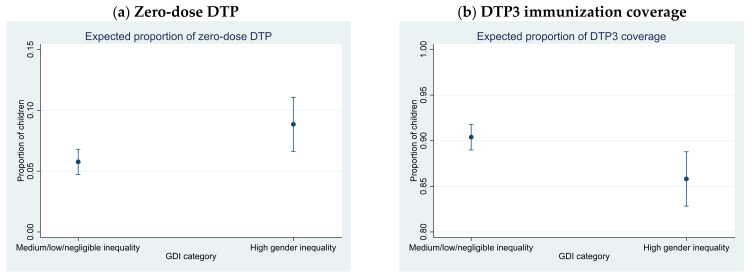
Adjusted proportions of (**a**) zero-dose DPT and (**b**) DTP3 immunization coverage for categories of GDI, 164 countries (1 country did not have available data for this model), 2010–2019. Note: The estimated proportions are adjusted for annual population growth and age structure (measured as the percentage of the population under 15 years of age), percentage of urban population, and the three individual dimensional indices of the HDI (health index, education index, and GNI index).

**Figure 2 vaccines-10-01032-f002:**
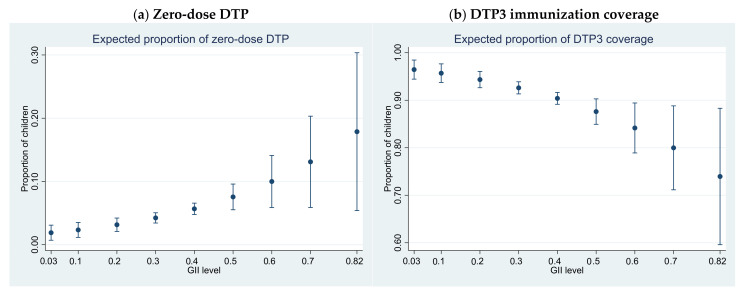
Average expected proportion of (**a**) zero-dose DTP and (**b**) DTP3 for fixed levels of GII, 161 countries (1 country did not have available data for this model), 2010–2019. Note: Results presented are from Model 2, controlling for annual population growth and age structure (measured as the percentage of the population under 15 years of age), percentage of urban population, and the three individual dimensional indices of the HDI (health index, education index, and GNI index).

**Table 1 vaccines-10-01032-t001:** Measures and data sources.

Category	Indicator	Source
**Outcomes**	Zero-dose DTP prevalence	Human Development Data Center [36]
	DTP3 immunization coverage	WHO Global Health Observatory [37]
**Gender inequality**	Gender development index (GDI)	Human Development Data Center [36]
	Gender inequality index (GII)	Human Development Data Center [36]
**Demographic** **characteristics**	Average annual rate of population change (%)	World Population Prospects [38]
	Population < 15 years (%)	World Population Prospects [38]
**Geographical context**	Urban population (%)	World Development Indicators [39]
**Human development**	Human development index (HDI) (0 to 1)	Human Development Data Center [36]
	Health index (0 to 1)	Global Data Lab [40]
	Education index (0 to 1)	Global Data Lab [40]
	Income index (0 to 1)	Global Data Lab [40]
	GDP per capita, PPP	World Development Indicators [39]
	Current health expenditure per capita, PPP	WHO Global Health Observatory [37]
	Mean years schooling population aged 25+	Global Data Lab [40]

**Table 2 vaccines-10-01032-t002:** Prevalence of zero-dose DTP and DTP3 immunization coverage by gender inequality, 2019.

		Zero-Dose DTP (%)				DTP3 Immunization Coverage (%)			
**Gender** **development** **index**		**Median**	**Min**	**Max**	**N**	**Median**	**Min**	**Max**	**N**
	High gender inequality	10.5	1	49	30	82.5	42	99	30
	Medium/low/negligible gender inequality	3	1	35	135	94	57	99	135
	*p*-value *	<0.001				<0.001			
**Gender** **inequality** **index**	High gender inequality	10	2	56	25	81	35	95	25
	Medium/low/negligible inequality	3	1	34	137	94	54	99	137
	*p*-value *	<0.001				<0.001			

* Test for equality of medians was carried out using quantile regression.

**Table 3 vaccines-10-01032-t003:** Odds ratios for zero dose DTP and DTP3 immunization coverage according to GDI category (up to 165 countries, 2010–2019).

	Model 1 (No Controls)	Model 2	Model 3	Model 4
**Zero-dose children**				
High gender inequality	3.651 ***	1.610 ***	1.560 **	1.688 ***
95% CI	2.51–5.31	1.13–2.30	1.10–2.20	1.14–2.51
**DTP3 immunization coverage**				
High gender inequality	0.278 ***	0.630 ***	0.639 ***	0.582 ***
95% CI	0.20–0.39	0.46–0.86	0.47–0.88	0.41–0.83
Number of observations	1628	1610	1618	1401

* *p* < 0.1; ** *p* < 0.05; *** *p* < 0.01. Note: Number of countries with available data for each model differs by included indicators. Not all countries had available data for all years.

**Table 4 vaccines-10-01032-t004:** Average marginal effects of GII on the predicted value of zero dose DTP and DTP3 immunization coverage, (up to 162 countries, 2010–2019).

	Model 1 (No Controls)	Model 2	Model 3	Model 4
**Zero-dose children**				
High gender inequality	0.208 ***	0.171 ***	0.169 ***	0.180 ***
95% CI	0.15–0.27	0.06–0.28	0.06–0.28	0.07–0.29
**DTP3 immunization coverage**				
High gender inequality	−0.324 ***	−0.251 ***	−0.250 ***	−0.295 ***
95% CI	−0.40–−0.25	−0.40–−0.10	−0.40–−0.10	−0.45–−0.14
Number of observations	1559	1541	1559	1343

* *p* < 0.1; ** *p* < 0.05; *** *p* < 0.01. Note: Number of countries with available data for each model differs by included indicators. Not all countries had available data for all years.

## Data Availability

Publicly available datasets were analyzed in this study. These data can be downloaded from the following locations: https://hdr.undp.org/en/data (accessed on 10 November 2021); https://www.who.int/data/gho (accessed on 1 June 2022); https://population.un.org/wpp/ (accessed on 1 June 2022); https://datatopics.worldbank.org/world-development-indicators/ (accessed on 1 June 2022); https://globaldatalab.org/ (accessed on 1 June 2022).

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
