# Peer review of "The Association between Childhood Immunization and Gender Inequality: A Multi-Country Ecological Analysis of Zero-Dose DTP Prevalence and DTP3 Immunization Coverage"

_vaccines, 2022, doi:10.3390/vaccines10071032_

Round 1

Reviewer 1 Report

Title: “The association between childhood immunization and gender inequality: a multi-country ecological analysis of zero-dose children and DTP3 immunization coverage.”

  • Please use “coverage” instead of “coverage.”
  • Please clarify: “a multi-country ecological analysis of zero-dose children”
  • Predictors of “childhood immunization”?

Abstract

  • “Gender inequality was positively and significantly associated with the proportion of zero-dose children (AOR=1.61, 95% CI: 1.13 - 2.30). Consistently, full DTP3 immunization was negatively and significantly associated with gender inequality (AOR=0.63, 95% CI: 0.46 - 0.86).” – “gender inequality” or “gender inequality index”?
  • What about “gender development index”?
  • “These associations were robust to the use of alternative gender inequality measures and model specifications controlling for demographic, economic, education, and health-related factors”; please clarify the meaning of this sentence.

Keywords: please consult MeSH terms: https://www.ncbi.nlm.nih.gov/mesh/; Ideally, please use some keywords that are not cited in title and/or abstract.

  1. Introduction

Line 33: “Gender equality is not only an important standalone goal, but also a key contributor 33 to and indicator of health of populations more broadly [1].” Why? Please give some examples/study findings.

  • Please cite some reviews on the present topic.
  • For instance see: https://scholar.google.com/scholar?start=10&q=%E2%80%9Cgender+inequality%E2%80%9D+and+immunization+review&hl=pt-PT&as_sdt=0,5

Line 56: “Childhood immunization can be captured by a number of indicators; we focus these analyses on two outcomes related to the combined diphtheria, tetanus and pertussis vaccine (DTP): (…)”. Please give more details about DTP? Mortality? Vaccination rates? When was this vaccine introduced? Vaccination Schedule? , etc.

  • Lines 55 vs. 56: please intelligibly link these two paragraphs.
  • Line 71: references are missing.
  • Please improve introduction.
  • Besides the study hypothesis, please present study objectives and the research question at the end of introduction.

  1. Materials and Methods

  • Methods are insufficiently described.

2.1. Indicators and data sources

  • Table 1: Column of source; please present a reference for all sources; please present a Link at least in the reference or in the reference and in the Table.
  • Table 1: Please define all terms when first presented in the text. For instance: Human development index (HDI) (0 to 1) and Human Development Data Center Health index (0 to 1) or Health index (0 to 1). What is the HDI? What is the Human Development Data Center Health index? Health index? , etc. For instance, present all definition in footnotes.
  • All methodological options must be justified. Why were the indicators of Table 1 selected (and not others)?

  • Immunization outcomes
  • Line 86: “based on the DTP vaccine” or “based on the number of administered vaccines” or other? Please carefully read all sentences.

2.1.2. Gender inequality

- Line 120: “We analyzed the GII as a continuous measure in regression analyses.” What about GDI?

- Line 197: “cross tabulations. (Table 2).” or “cross tabulations (Table 2).” Please carefully proofread the paper.

2.2. Analyses

- How were the models validated? What were the validation measures? (Cite some references) Please describe the validation measures in section2.2. Please present the validation measures in results.

Results

  • References are missing in results. Please check.
  • 1. Descriptive analyses: It seems the presentation of some results is missing (or not?). Please present the results from all indicators (see Table 1).
  • Please try to improve the presentation/comprehensibility of study findings.

Discussion

  • Please cite more studies/reviews in discussion. Please note that authors only have cited 4 references in discussion, which is clearly insufficient.
  • Please check if the discussion follows the critical explanation/discussion of the following topics: a) Descriptive analyses; Regression analyses: b) Gender development index and c) Gender inequality index. Please ensure that all study findings are exhaustively discussed. d) The study hypothesis and research questions should be discussed in individual paragraphs.
  • Please define at end the discussion the following sections: study strengths, study weakness, possible sources of study biases, practical implications, and future research.
  • If necessary, authors can create subheadings to organize/structure the discussion.

Conclusion

  • Please present a brief conclusion. Please write at least a sentence per each defined study objective.

References

  • Please check the format of all references. For instance, reference 12…
  • “Martin Hilber, A.; Bosch-Capblanch, X.; Schindler, C.; Beck, L.; Sécula, F.; McKenzie, O.; Gari, S.; Stuckli, C.; Merten, S. 389 Gender and immunisation: summary report for SAGE, November 2010. 2010.”
  • Please carefully read instructions for authors.

Author Response

We thank Reviewer 1 for their review, and appreciate the opportunity to strengthen the manuscript with this feedback.

Reviewer 2 Report

Dear Authors,

Thank you for your submitted work, which I am delighted to read. The paper has been written clearly and provided sufficient background. The results provided by the research follow the data and analysis logically.

It is also a plus that the paper has provided a useful statement on the limitations of the present study.

I have only two comments. In the final discussion part, the authors should spend space providing the audience with their conclusions about the possibility of attaining better equality, and if the available evidence could help address the future issues of policy failures. Please refer to the view of cost of science and risks of policy failures provided here: https://www.nature.com/articles/s41599-022-01034-6

Second, perhaps the authors could also discuss the issues of cost concerning health services. This is one of the most burdensome factors for worldwide populations.

Otherwise, the paper is in good shape.

Wishing you all the best.

Author Response

We thank Reviewer 1 for their review, and appreciate the opportunity to strengthen the manuscript with this feedback.

We thank Reviewer 2 for their review and insight, and appreciate the opportunity to strengthen the manuscript, particularly the discussion, with this feedback.

Dear Authors,

Thank you for your submitted work, which I am delighted to read. The paper has been written clearly and provided sufficient background. The results provided by the research follow the data and analysis logically.

It is also a plus that the paper has provided a useful statement on the limitations of the present study.

I have only two comments.

  • In the final discussion part, the authors should spend space providing the audience with their conclusions about the possibility of attaining better equality, and if the available evidence could help address the future issues of policy failures. Please refer to the view of cost of science and risks of policy failures provided here: https://www.nature.com/articles/s41599-022-01034-6

We have added additional detail regarding practical suggestions/implications of efforts to improve gender equity and therefore improve immunization equity as well (see revised Discussion Paragraph 3); the availability of such policy and practice changes, in our opinion, makes the achievement of greater equality possible. I believe the issue of policy failures is somewhat outside the scope of our analysis and expertise, however, the linked article is of course an important and relevant consideration for public health efforts. We do now suggest that future research involve rigorous evaluation of interventions to improve gender equity as a mechanism to increase immunization coverage, to demonstrate effectiveness and feasibility prior to any widespread policy implementation.

  • Second, perhaps the authors could also discuss the issues of cost concerning health services. This is one of the most burdensome factors for worldwide populations.

We have added to the discussion brief mention of the cost-effectiveness of gender equity focused immunization interventions: “While childhood immunization generally is highly cost-effective, multi-sectoral approaches will likely not be the most cost-effective interventions short term. However, systemic approaches such as these have greater potential sustained and wide-ranging impact, and key stakeholders in immunization will need to consider novel approaches such as these to vaccinate the hardest-to-reach populations in the effort to leave no one behind” (lines 364-369). Unfortunately, we could find no published literature to date examining the cost of gender-transformative interventions in childhood immunization; we therefore suggest the evaluation of such interventions as a key area of future research to determine feasibility and cost effectiveness: “Future research should in particular implement and evaluate interventions which target gender inequality as a mechanism to improve immunization coverage, to better determine the feasibility, costs, and practical impact of gender-transformative interventions in this space” (lines 397-400).

Otherwise, the paper is in good shape.

Reviewer 3 Report

The paper uses the gender identity theory to explain the overall vaccination rate. I am not sure this is feasible to state in that way, as this is likely not a causal relationship. Although it may be associated, and the data shows it is, I am unsure how to handle this apparent lack of causality. This would require a more robust model, where it could be tested somehow if these two have a causal relationship. Nevertheless, if the authors manage to describe and adjust this correctly, I think the association they found is worth looking into in more detail. However, I fear that aiming for gender-based policies during the epidemic is not the best way forward and that it would be best to downplay this association slightly. There is likely a hidden layer (the latent class concept), where the generalized health literacy underlines both of the measured variables.

The second problem is the ecological fallacy, where the group-based conclusions may not entirely reflect the processes at the individual level. I wonder could you maybe make an interaction item for regression and try to assess the variance explained by various factors.

An additional layer of support might come from the yearly analysis, with the opportunity to observe trends in the development – do they follow the trends in vaccination? If yes, you might have a stronger case exhibiting causality (please note that the causal association is one worth developing an intervention for, while the association may not be usable at all).

Author Response

We thank Reviewer 3 for their review, and appreciate the opportunity to strengthen the manuscript with this feedback.

Round 2

Reviewer 1 Report

Dear authors, Thanks for all updates. Congratulations the quality of the paper has been improved.

 Some minor comments:

 Introduction

“Only one study to date has examined gender inequality  and childhood immunization coverage at the national level across countries, and found  that greater gender inequality was associated with lower and less equitable immunization coverage [17]”.

-          Date/timeframe and number of countries covered in this study [17]?

Discussion

-          Please discuss the differences and similarities between the findings of the present study and reference 17.

-          Please discuss how the present findings can be used in campaigns, political health measures or educational interventions. Authors may create a section on practical implications at the end of introduction.

Author Response

We thank Reviewer 1 for their additional review!

Introduction

“Only one study to date has examined gender inequality and childhood immunization coverage at the national level across countries, and found that greater gender inequality was associated with lower and less equitable immunization coverage [17]”.

  • Date/timeframe and number of countries covered in this study [17]?

This information has been added in text: “Only one study to date has examined gender inequality and childhood immunization coverage at the national level across 45 countries using data from 2005-2014, and found that greater gender inequality was associated with lower and less equitable immunization coverage [17].”

Discussion

  • Please discuss the differences and similarities between the findings of the present study and reference 17.

This information has been added in text: “Results from this study align with a previous national-level analysis [17], despite a larger and more global sample (165 rather than 45 countries), more recent data (2010-2019 rather than 2005-2014), multiple years of data for countries (rather than a single most recent survey), and an alternate analysis approach (fractional logistic regression rather than a meta-regression approach). Robustness of the findings despite these differences strengthens the conclusions presented here.”

  • Please discuss how the present findings can be used in campaigns, political health measures or educational interventions. Authors may create a section on practical implications at the end of introduction.

We have elected to include this information in the discussion rather than introduction to speak to the observed findings (rather than hypothetical findings, as would need to be presented in the introduction). We have expanded discussion paragraph 3 with additional examples of practical implications and suggested approaches/interventions/measures. We also clarify that the suggested gender-responsive measures would themselves be practical responses to the findings presented:

Findings from this study suggest that policies and interventions which directly address gender inequality are needed to ensure high and equitable immunization coverage. Gender responsive measures in policy and practice can address some of the immediate challenges faced by caregivers in accessing immunization services for their children. Practical examples of such measures include establishing the location and timing of vaccine clinics based on mothers’ work schedules and time availability, holding vaccination clinics in locations that are easily accessible for mothers, or having female vaccinators in communities where it is not socially acceptable for men and women to interact. These measures can improve coverage services for a short period; however, approaches that have sustained impact will be those that address deeper in-equalities, including those factors captured by the GDI and GII such as education and control of resources. Maternal education, and by proxy, gender inequality in education, has in particular been extensively studied and shown to be a determinant of child immunization [46-50]. Gender transformative approaches therefore must focus on shifting power imbalances and addressing social norms, beliefs and attitudes which create and sustain discriminatory policies and practices across sectors. Gender-transformative approaches can include supporting interventions that distribute household responsibilities among both parents, supporting fathers to be engaged in child care and seeking health services for children, ensuring equitable payments among health professionals, ensuring equitable access to and utilization of educational opportunities, ensuring equal representation of women and men in decision making positions, and prevention of sexual assault and harassment in the health sector [10,51].

Reviewer 2 Report

Dear authors,

Thank you for your revised manuscript.

Having examined it again, I found the paper addressing the remaining points satisfactorily.

Best regards,

Author Response

We thank reviewer 2 for their additional review.

Reviewer 3 Report

Improved as far as possible

Author Response

We thank reviewer 3 for their additional review.